*Data & Policy* (2021), 3: e33

CAMBRIDGE
UNIVERSITY PRESS

**TRANSLATIONAL ARTICLE**

# Data sharing and collaborations with Telco data during the COVID-19 pandemic: A Vodafone case study

Pedro Rente Lourenco* ⓘ, Gurjeet Kaur, Matthew Allison and Terry Evetts

Big Data and Artificial Intelligence, Vodafone Group, London, United Kingdom
*Corresponding author. E-mail: lrl.pedro@gmail.com

**Key words:** COVID-19; data ethics; data privacy; data sharing and collaborations; mobility insights

**Abbreviations:** CDRs call detail records; KPI key performance indicator; OD origin destination; ROG Radius of Gyration

**Abstract**

With the outbreak of COVID-19 across Europe, anonymized telecommunications data provides a key insight into population level mobility and assessing the impact and effectiveness of containment measures. Vodafone's response across its global footprint was fast and delivered key new metrics for the pandemic that have proven to be useful for a number of external entities. Cooperation with national governments and supra-national entities to help fight the COVID-19 pandemic was a key part of Vodafone's response, and in this article the different methodologies developed are analyzed, as well as the key collaborations established in this context. In this article we also analyze the regulatory challenges found, and how these can pose a risk of the full benefits of these insights not being harnessed, despite clear and efficient Privacy and Ethics assessments to ensure individual safety and data privacy.

**Policy Significance Statement**

Vodafone encourages sharing and reuse of data through voluntary, market-driven mechanisms. Data sharing should only take place where it is legally compliant, ethical and socially acceptable, in line with the principles of trustworthiness and privacy-by-design. To be sustainable, sharing and reuse of data should be subject to fair remuneration that recognizes the significant investment required to collect, curate and maintain data, as well as produce meaningful and accurate insights. It is of particular importance that policy fosters innovation through data, and this can only exist in a sustainable market for data analytics and insights, that levels the playing field across the world and between the different players, and that at the same time extracts real value for society.

## 1. Introduction

Anonymized and aggregated telecommunications data of cell tower locations ("Telco Data") offers a wealth of possibilities for innovation and impact in society, particularly when combined with other data sources and shared with entities whose purpose is to have meaningful positive impact in people's lives. There are many examples of the value to be extracted from these datasets, as they offer a clear proxy of human activity that can be used to map poverty, design containment and mitigation strategies for epidemics or advise on urban mobility (Steenbruggen et al., 2013) and human behavior (Douglass et al., 2015), among others.

In the process of using Telco data it is imperative to ensure privacy standards are met, and that proper ethical assessments are conducted so that a trusting relationship can be established between the individuals and the company that holds their data. By implementing privacy controls such as anonymization and aggregation, the benefits of analyzing these datasets can be unleashed while preserving the user's privacy. Furthermore, the risk of unintended consequences against individuals and society must be assessed and mitigated for example insights on minority population movements which will be discussed in Section 2.3.

Telecommunication networks have different architectures and the backend systems collect different types of data, depending on the operating company. This means that an additional challenge is posed when the intention is to make cross-country and cross-operator analyses.

The COVID-19 pandemic hit Europe in March 2020, with quick spread from Italy to the whole of Europe and with devastating consequences for society. Being a human-mediated transmission type of disease, human behavior was an obvious factor to take into account in exit strategies for the pandemic and containment strategies for emergency situations across the world. Telco data, even at its most basic level, offers the possibility of extracting value to analyze population-level behaviors and patterns of mobility. As telecommunication networks depend on physical "cell sites"—places where the radio transmitters are placed to offer network coverage, and the connections to these have to be monitored by the network—for quality purposes, billing and operations—an approximate view of a cell phone's location can be inferred. While cell based location today is still less accurate than other methods (such as GPS), we recognize it may reveal sensitive information about an individual and we had to define a more elaborate privacy by design approach; this entailed only collecting location relevant information for analysis, replacing all user IDs with randomly generated IDs (pseudonymisation), as well as aggregating to the level of 50 users or more and overlaying information on large geographical areas.

## 2. Methodology

With approximate location data and relatively high sampling frequencies on customer bases of several million cell phones, the representativeness of the data for mobility insights is very high, particularly in countries where cell phones are widespread and the usage is very high. The ubiquitous nature of these devices makes them uniquely positioned to offer important insights into human mobility and, consequently, into disease transmission patterns.

In order to have results comparable across countries and due to the differences in raw data capture methodologies, techniques were developed to allow a common framework across the Vodafone footprint, using different kinds of data. This study will focus on two types of analysis, and three types of datasets.

### 2.1.  Raw datasets

Across the Vodafone footprint three different types of datasets were analyzed: Call Detail Records (CDRs)/x-Data Records (XDRs), Probe data, and App-based data.

CDRs and XDRs are datasets that capture only active events in the network. These are records of when a customer sends or receives a voice call, text message, multimedia message, or data packet. This means that the sampling frequency depends heavily on the usage of the cell phone, likely to be higher in societies where the usage of baseline services depends on data (such as web-based messaging or social media running in the background). XDRs tend to have higher sampling frequencies as they are exchanging data packets through the network, whereas CDRs typically correspond to voice and SMS communications. From these datasets, we used only the cell locations and timestamps.

Probe data is captured passively, often for traffic management and network operations. Each cell phone exchanges information with the network beyond active events and these signals represent an approximate location—just like with CDRs. These are higher frequency signals but they are not available at the same sampling rate across all networks, and not all operators capture these signals for analysis beyond network operations.

App-based data (Wang et al., 2019) refers to data captured from a specific app, in this case the "myVodafone" app, in which customers consent to capturing of multiple signals from their phones,

including networks speeds to monitor network quality. Some of these signals come with a cell tower position as well, which makes them somewhat equivalent to the other two datasets. The cohort of customers in this case is more limited as only app users who have consented to the use of their data will be providing information.

Data was captured from the start of February 2020 in order to capture pre-pandemic behaviors. All of the data is automatically generated by the networks and in this study we are reusing these datasets to provide anonymous analytics which are compatible with the original purpose of the data collection.

## 2.2. Types of analysis

In this work, two types of insights were extracted: origin-destination (OD) matrices and behavioral KPIs.

### 2.2.1. Home locations

Approximate home locations—or home "cell" locations—were calculated for each day by looking at night-time events (Colak et al., 2015). Given the nature of the measures implemented by governments it was decided not to have a longer estimation for home locations for each individual, because of changes in location pre and post lockdown, with large exodus from urban centers into second homes and family homes in more rural areas. The top 3 night cells the cell phone connects to be extracted, and "night" was defined as the period between 22:00 and 5:00, local time.

### 2.2.2. OD matrices

OD matrices allow an analysis of mobility across a country. By splitting a country into smaller regions (NUTS3 in Europe, local units in other countries) (Eurostat (European Commission), 2020) and counting the numbers of movements between the regions in each day, matrices were extracted that represent the estimated number of people moving from one region to another on each day. Boundary effects were negligible as the in-country geographical splits were large enough to ignore the small amounts of cell towers at the borders of each region that could overlap with other cells and cause odd "back and forth" behaviors.

These OD matrices were also scaled to population levels, based on home locations from the previous night and Eurostat population estimates for NUTS3 regions. By calculating the ratio of customers located in a certain region and dividing the population count by this value we get a multiplier to extrapolate numbers of customers moving to numbers of people moving.

Population estimates from Eurostat are updated as new information is collected for each country, and for the purposes of this study the most recent updates were used for each region. Population under 18 years of age is not counted in this dataset and is assumed to follow a similar behavior to the rest of the population. In order to take into account nuances in the market share in every region, the population-level scaling was done per NUTS 3 region. We assumed the demographic profile of Vodafone's customer base to not be significantly different to that of the wider population.

### 2.2.3. Behavioral key performance indicator (KPI): Percentage of time at home cell

Using the approximate home cell calculations and considering the three potential home cells for each customer, the study processed the time in these home cells and the time out of these home cells to extract this KPI. The equation below reflects its calculation:

$$\tau_{home} = \frac{t_{in}}{t_{total}} \tag{1}$$

where $t_{in}$ is the time the customer is seen at his home cells and $t_{total}$ is the total time the customer is seen in the network.

### 2.2.4. Behavioral KPI: Radius of Gyration

González et al. (2008) defined the Radius of Gyration for human mobility in terms of its center of mass. In this study we use an adapted measure taking into account the most common home cell as the reference (i.e., the cell to which a customer is connected to the most during the night hours). Using these latitude and longitude values, as per the equations below, we can calculate the Haversine distance (Winarno et al., 2017)—the distance between the coordinates of all cell towers visited by the customer (independently of how many times these were visited) and the top home cell. From this we can calculate the Radius of Gyration ("ROG").

$$a = \sin^2\left(\frac{\Delta lat}{2}\right) + \cos(lon_1) \cdot \cos(lon_2) \cdot \sin^2\left(\frac{\Delta lon}{2}\right)$$
$$c = 2 \cdot a \tan 2\left(\sqrt{a}\sqrt{1-a}\right)$$
$$distance = R \times c$$

(2)

where $R$ is the approximate radius of Earth (3,671 km). We then calculate ROG with

$$ROG = \sqrt{\frac{1}{n_{cells}}\sum_{i=1}^{n}(c_i - c_{home})^2}$$

(3)

where $(c_i - c_{home})$ is given by the Haversine distance calculated above.

The Radius of Gyration was calculated daily to allow a more granular insight into the behavioral changes through time.

### 2.2.5. Behavioral KPI: Collocation

As COVID-19 transmission is highly influenced by the numbers of people aggregating in the same space and time, a KPI of co-location was used—to assess the amount of people aggregated in a place (different from their home locations) at the same time.

For this 60 min buckets were processed throughout the day and looked into the numbers of customers seen in a specific cell in the network at the same time, as long as this was not their home cell (where it was assumed they would be in their houses). This assumption is quite robust in densely populated areas but can have some shortfalls in areas where the distribution of cell towers is more scarce. We address these limitations in a dedicates section further on.

### 2.2.6. Behavioral KPI: Number of cells visited

Another metric of the amount of mobility a customer has in the network is the diversity of cell towers "visited" per day, this was calculated to assess the levels of mobility restrictions pre and post lockdown measures were imposed.

### 2.3. Safeguarding individual privacy

In the process of acquiring mobility insights it is crucial that privacy and ethics are addressed from a design phase also known as "Privacy by Design."

A Privacy Impact Assessment was conducted to determine the requirements needed to mitigate the privacy risks. This Assessment was conducted using a privacy assessment tool containing 72 questions around the types of data to be used, the purposes, the location of processing and the privacy measures applied to the processing. Separately, a security assessment was conducted to ensure that data was processed in a secure encrypted manner.

The privacy requirements required all insights to be aggregated at a level of 50 or more individuals; this means that during aggregation only movements of 50 or more individuals are

represented, and groups below this threshold are discarded and represented as missing data. Furthermore only location network data was used, which is processed for lawful purposes within agreed retention periods; this data was pseudonymized with unique random user IDs and a country by country compliance analysis was performed, and the analysis was not performed if not agreeable by each regulatory body.

In order to get to the anonymized and aggregated tables seen in the figures, journeys need to be mapped. As an additional protection, the internal analysis is done on pseudonymized data sets to reduce the risk of analysts being able to directly identify individual journeys while still allowing the journey data to be analyzed. Once the data has been collated, the anonymized and aggregated outputs were shared with external parties, no personal data or unique identifiers were shared outside of the organization. Each data sharing activity—external to the company—undergoes its own assessment. This assessment analyses whether the external party is "legitimate" in their use of the data and it is not being used for negative purposes. They are also asked to sign up to contractual obligations prohibiting them from misusing the data or attempting to mix the data with other data with an aim to identify individuals. The risk of this being successful was low but protections were sought nevertheless.

### 2.4. Ethical assessments

Not only did we require controls to ensure individual privacy, we were also aware of the risks the insights might pose to minority group movements. For example, if a particular minority group resided in one area, would the insights show the movements of that minority group? Those insights could then be used by certain parties who might target those minority groups.

In order to mitigate these risks, we completed a "Group Human Impact Assessment", which went further than assessing the individual right to privacy but the privacy of Groups or demographics as a whole. A country-by-country assessment was completed for the markets where we generated the insights. We then used external human rights benchmarking, European Court of Human Rights jurisprudence and reviewed that country's surveillance regime to conclude what minority groups may be at risk. Once identified, we tested the data to see if movements in minority group areas could be gleaned. If so, the aggregation was increased to remove that risk. If the risk was still present or if there would be a plausible risk of data being misused, then data was not shared.

## 3. Results

### 3.1. Behavioral KPIs

In Figures 5 and 6, the different behavioral KPIs can be seen, plotted as the perceptual difference from a baseline set as the average KPI value for the first week of February. This was considered as a baseline mobility level as the pandemic had not hit these countries at that point, and it allows us to have a comparative analysis of the evolution of mobility and its reduction across different countries.

Figure 7 shows the evolution of reported COVID-19 cases (right axis) and the same for mobility metrics for some example countries.

## 4. Collaboration and Data Sharing

### 4.1. Principles for data sharing

No personal data from mobility insights was ever shared. Data is processed internally with pseudonymized customer identifiers and within secure servers and cloud environments throughout the Vodafone systems and footprint. Several security assessments were undertaken to ensure privacy and security concerns were addressed.

Furthermore, strict Terms and Conditions were set for the usage of Vodafone platforms that allow access to data, and this access was only granted upon a clear Privacy Impact Assessment and agreement of

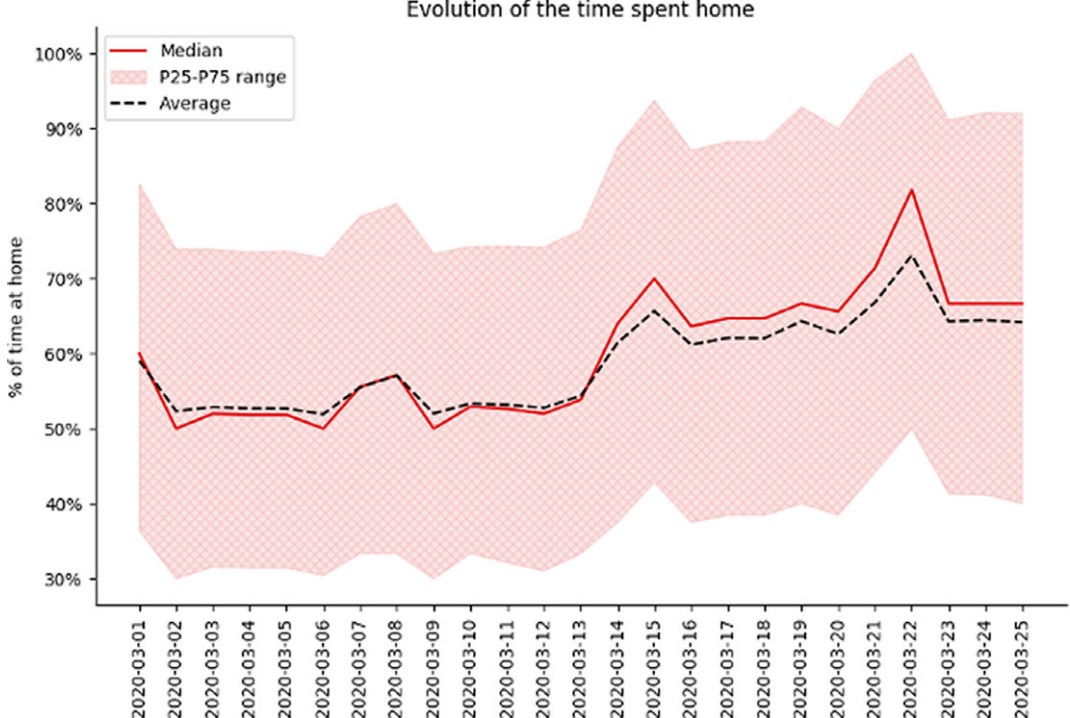

**Figure 1.** *Example for Spain on the evolution of time spent at home at the beginning of the epidemic. We can clearly see an increase in the time spent at home throughout the country after lockdown measures were implemented (on March 14, 2020).*

third parties to all of Vodafone's terms. The mobility insights shared were on a strict case by case basis and only for the purposes agreed, with accesses revoked as soon as these expired or were no longer relevant.

### 4.2. Regulatory challenges and frameworks

The sensitivity of mobile location data is very important for Vodafone, and the needs to ensure full anonymization of this data are very well understood. In all data sharing events, Vodafone only ever provides fully anonymous, aggregated insights, in order to prevent reidentification of customers. Across Vodafone's footprint, a number of different regulations apply, and although there have been initiatives to converge these into common ground in some geographical areas (e.g., GDPR or the ePrivacy directive), this is a somewhat localized effort, and it has suffered from fragmentation in the application of such directives across countries. Some data protection regulators (Wood, 2020) shared their support for these types of analytics where others did not. This in turn causes uncertainty and slows down the usage of valuable insights like the ones presented in this paper. This is even clearer across other geographical areas, where the diversity of regulations, or the lack thereof, make it very difficult for data collaboratives to exist and thrive (Wiewórowski, 2020).

Vodafone encourages data sharing initiatives through voluntary, market-driven mechanisms, in situations where it is legally compliant, ethical and socially acceptable, in line with the principles of trustworthiness and privacy-by-design. In order to maintain sustainability, these should be subject to fair remuneration that recognizes the significant investment required to collect, curate and maintain data, as well as produce meaningful and accurate insights. It is of particular importance that policy fosters innovation through data, and this can only exist in a sustainable market for data, that levels the playing field across the world and between the different players, and that at the same time extracts real value for society.

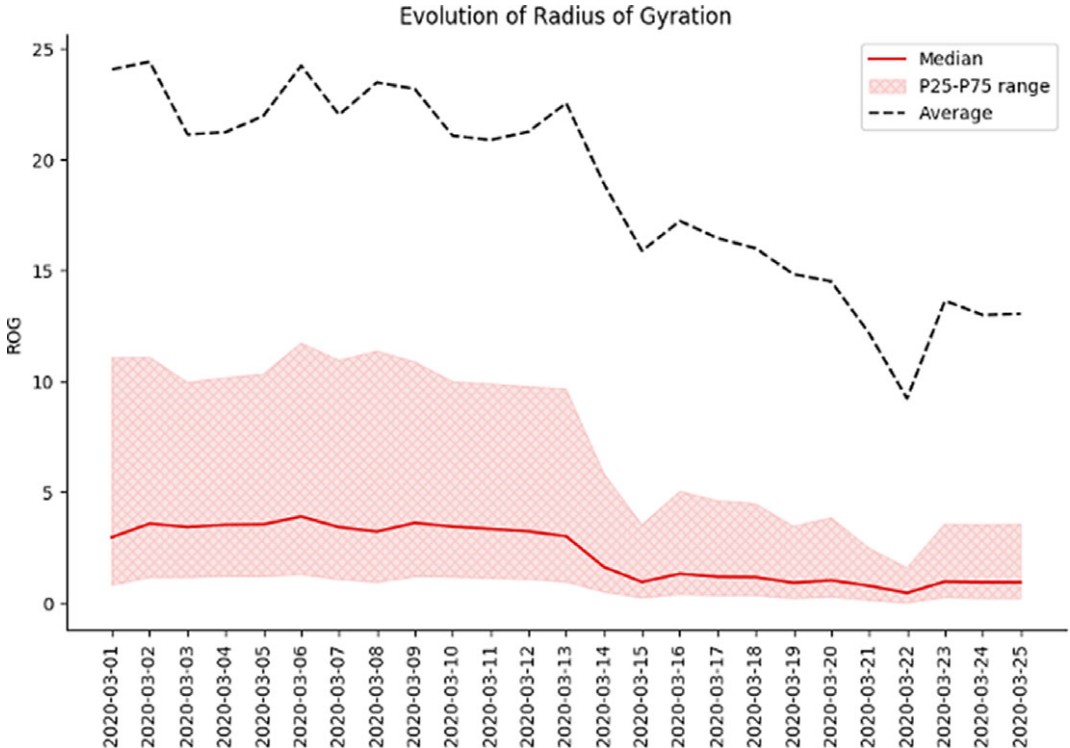

***Figure 2.*** *Example for Spain on the evolution of the Radius of Gyration at the beginning of the epidemic. We can clearly see a decrease in the ROG throughout the country after lockdown measures were implemented (on March 14, 2020). We can also notice some large outliers are present in the data as the average is much higher than the median and P25–75 range.*

### 4.3. Data usage and usefulness

Within the context of this COVID-19 analytics project, Vodafone has shared anonymized and aggregated insights with the European Commission's Joint Research Council (JRC), the International Monetary Fund (IMF), the World Bank (WB) and the United Nations Children's Fund (UNICEF). These entities have been working in close collaboration with Vodafone to extract value from these datasets for different purposes, which we state below.

The JRC is working across Europe with multiple Telco Operators, in order to have a global view of mobility across the continent and how this has been impacted by the pandemic, as well as how mobility is impacting the spread of COVID-19, by extracting relevant network metrics from OD matrices and tracking them across time. The initial reports can be found online (Iacus et al., 2020a, b; Santamaria et al., 2020).

With the IMF, Vodafone has been working closely to understand disparities of the impact of lockdowns in different age groups and genders. By disaggregating the self-isolation KPIs developed into different age groups and genders, we uncovered different levels of impact for different population segments, which allows for better policy decisions to fight inequality. This research has been conducted across European countries where Vodafone has a large presence (namely Portugal, Italy, and Spain) and is published in the World Economic Outlook of 2020; this work has also been conducted in South Africa, where income gaps are a source of inequality and is currently ongoing.

The purpose of the cooperation with the World Bank is so this entity can use Vodafone's insights in their policy advice and economic modeling for governments around the world.

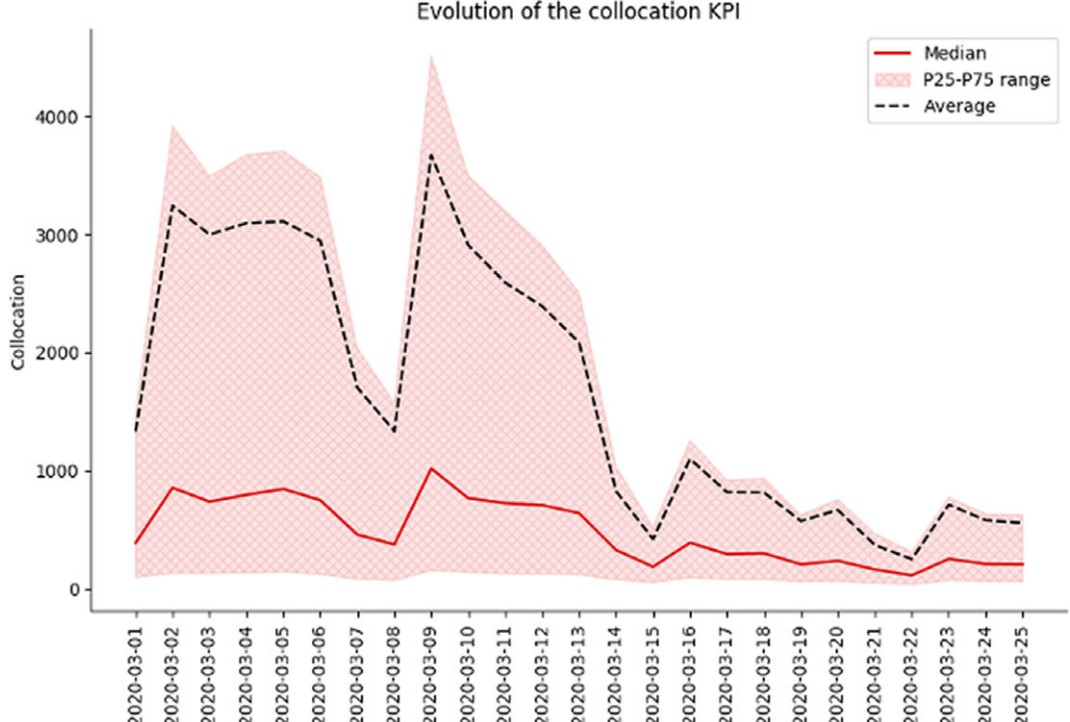

***Figure 3.** Example for Spain on the evolution of the collocation KPI at the beginning of the epidemic. We can clearly see an decrease in the collocation KPI throughout the country after lockdown measures were implemented (on March 14, 2020). The dip in this KPI on March 8, 2020 corresponds to a set of missing data for that period due to technical error.*

UNICEF has expressed interest in evaluating regional differences in these KPIs, as these can uncover social inequalities that put certain population segments at risk. This is particularly important to evaluate the income inequalities and has been a target of UNICEF's research since the start of the pandemic (Garcia-Herranz et al., 2020). More particularly UNICEF's Country Offices have been using mobility insights to assess the risk and impact of schools reopening, mitigating the impact of social distancing measures, use mobility in epidemiological modeling of the disease and consider socio-economic and urban/rural differences in mobility reduction to better tailor mitigation policies.

Through a collaboration with the WorldPop project at the University of Southampton, Vodafone has used these anonymized and aggregated insights to inform epidemiological models and insights (Floyd et al., 2020). The geographical spread of disease is a key factor to take into account in policy matters, and the study performed across Europe revealed that there is value in coordinated containment measures across countries to slow down the spread of COVID-19 and the rates of infection (Ruktanonchai et al., 2020).

## 5. Discussion

### 5.1. Representativeness and limitations

Telco datasets offer a highly representative overview of population behaviors, due to their large scope, having lower rates of fake accounts and high frequency of data. Furthermore, they overcome the "smartphone-only" limitations of app-based location tracking services, which makes them more inclusive and less prone to biases like income gaps and generational bias.

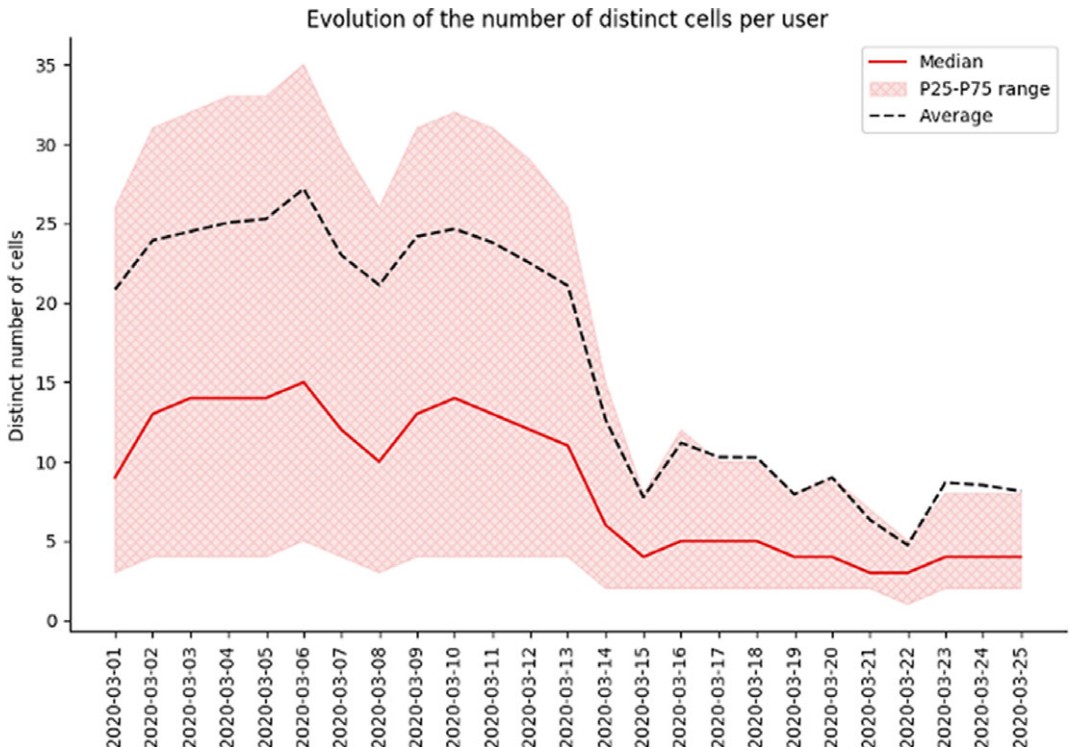

**Figure 4.** *Example for Spain on the evolution of time spent at home at the beginning of the epidemic. We can clearly see a decrease in the number of distinct cells visited throughout the country after lockdown measures were implemented (on March 14, 2020).*

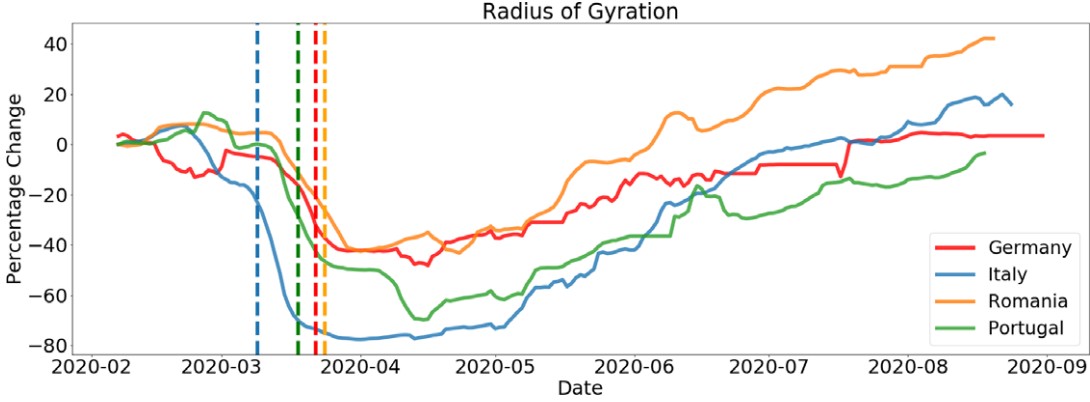

**Figure 5.** *Average Radius of Gyration for several European countries. Plots show a 7-day moving average with the respected lockdown date represented by a dotted vertical line. We see the fast drop once lockdown measures are implemented and a slow recovery of mobility as the pandemic evolves.*

However, these datasets are limited to the density of the Telco networks, which depend on population density and the company's own strategy and technologies. Geographical locations with higher population densities, such as large metropoles, will offer higher sensitivity to movement, as mobile phones will connect to many different Base Transmitter Stations (BTS) throughout the day, even with short ranged movements, in order to offer the best service in a busy area of the network. On the other hand,

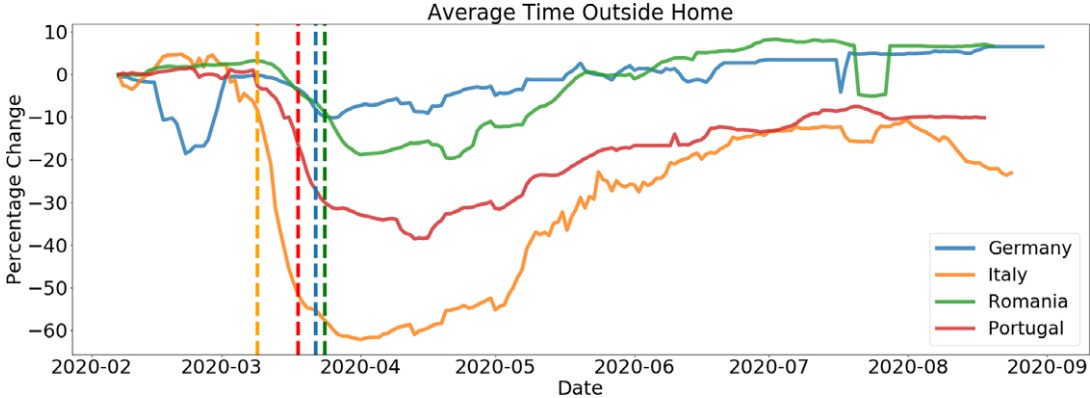

**Figure 6.** *Average time outside home for several European countries. Plots show a 7-day moving average with the respected lockdown date represented by a dotted vertical line. We see the fast drop once lockdown measures are implemented and a slow recovery of mobility as the pandemic evolves.*

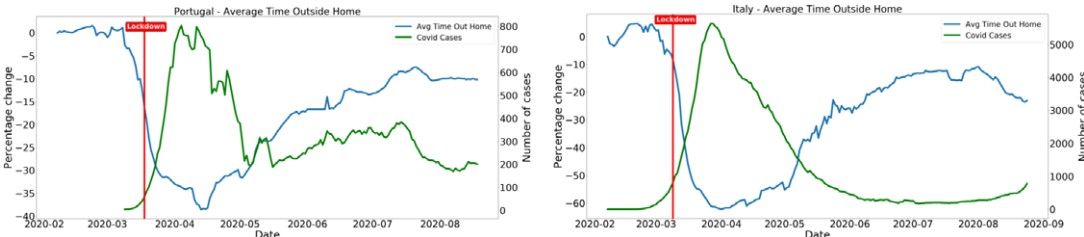

**Figure 7.** *Evolution of Average time outside home (left axis) and daily reported COVID-19 cases. Left plot refers to Portugal and the right plot to Italy. We can see that although there is a clear inverse correlation between the number of daily cases and mobility, other underlying factors contribute to the spread of the epidemic, as the number of daily cases in Portugal goes up as mobility increases and is then lowered, due to other measures being taken in the country. The same does not seem to happen for Italy.*

geographical locations with lower population densities, such as rural areas, will see less traffic in the network and will not need as many BTS units, which make the coverage much wider, and the inference of locations less precise. This affects many of the calculations, as for example the home cell determination uses a specific number of cells to be considered during the night (in this case 3) and it is much easier for a mobile phone to connect to more than 3 different cells during night hours in a city than it is in a rural area. Our aggregation rules also mean that less populated areas would not be represented.

Despite these limitations, anonymized and aggregated Telco data still offers an extremely valuable set of insights on general population mobility and can be a key element in social good research, as has been thoroughly shown during the COVID-19 pandemic. There is potential for near real-time insight which can be extremely useful for government bodies and other stakeholders, and despite the large amount of compute effort and the need for close monitoring of the data and insights, this is exactly what was produced by Vodafone in this study. By providing a platform with near real-time anonymized insights we aimed at making the most useful information available to the right people at the right time, in contrast to the more usual historical analysis approach seen in this kind of study.

**Acknowledgments.** We are grateful for the technical assistance of A. Marin, D. Patané, M. Dziduch, M. Pirvulescu, and S. Foster from the Vodafone Group Big Data and Artificial Intelligence team, as well as D. Gonzalez, D. Fierro, G. Lastra, and L. Rodriguez Solis from Vodafone Business in the development of the KPIs. We would also like to thank the Big Data and Artificial Intelligence teams in all of the participating Vodafone local markets.

**Supplementary Materials.** To view supplementary material for this article, please visit http://dx.doi.org/10.1017/dap.2021.26.

**Funding Statement.** This research was supported by Vodafone Group Plc.

**Competing Interests.** The authors declare no competing interests exist.

**Author Contributions.** Conceptualization: P.R.L.; Methodology: P.R.L.; Data curation: P.R.L.; Data visualization: P.R.L., G.K.; Writing original draft: P.R.L. All authors approved the final submitted draft.

**Data Availability Statement.** Data and code are only shared on a strictly case by case basis as per Vodafone's Privacy and Security standards.

**Ethical Standards.** The research meets all ethical guidelines, including adherence to the legal requirements of the study country.

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
