## [Reviewer Report]

This work represents a summary of a global effort within Vodafone Group to tackle COVID-19 and provide relevant insights to institutions and policymakers. At a time of great changes in population behaviour, the insights provided by Vodafone have proven to be extremely relevant for many organisations and the impact these types of insights can have in policy making are great. At the same time and at the forefront of all of Vodafone's activities the ethical and privacy questions were addressed to ensure safety and trust from the public. We hope this paper sheds a light into how important these activities are and elicits more collaboration across the industry and public organisations to address societal problems.

---

## [Reviewer Report]

*Comments to Author*: Thank you for the opportunity to review this interesting research article. In general, the article provides practical and detailed insights into deriving movement patterns from telecommunications data, and into the use of this data in the fight against Covid-19. This article adds value to the current discussion on mobile data analytics through a solid and comprehensive case study. The authors explain how mobility patterns are derived from telecommunications data, what kind of analyses Vodafone was able to make from the data during the pandemic, and describe the challenges and considerations related to telco data analytics.

However, there are some opportunities for revision that would make the article even more valuable and I would like to offer my suggestions on how to improve it.

While the article makes a good case study, it could have a broader perspective. The article sets Vodafone’s analytics into the context of Covid-19 and explains the benefits of telco data in assessing the effectiveness of restrictions measures in several countries. In addition, the authors touch on many limitations of telco data throughout the article. However, the article could address these limitations more clearly, while providing more background to the choices made during the project.

Structurally, the article would benefit from the introduction of a section dealing with the general factors and limitations affecting telco data analytics. Throughout the article, the authors present important considerations about, among others, legal restrictions for telco data analytics in the EU, extrapolating operator data to the population level, and the effects of population density to the available data. I would suggest that a section be added to the beginning of the article, explaining all factors that are essential to understanding the benefits and limitations of the data and that help justify some of the choices made in terms of data accuracy and availability.

Furthermore, the article excellently presents the valuable information and unique insights telco data can bring into policymaking through accurate and up-to-date movement analysis. However, I believe that the article could contribute more to both scientific and political discussion by also addressing the loss of accuracy resulting from the aggregation and anonymization of data. I fully agree with the authors that these measures are indispensable to preserve the privacy of mobile customers and also the ethicality of this kind of analysis. At the same time, the balance between accuracy and full anonymization is one of the most difficult questions in telco data analytics, and the article has great potential to contribute more to this discussion. The decisions made in this case to limit group sizes to a minimum of 50 and geographical areas to NUTS3 seem reasonable but it would be interesting to hear about the rationale behind these choices, and the arguments considered in determining these limits.

In addition to my general remarks above, I hope that the following suggestions, albeit less fundamental and more detailed by nature, will also prove helpful:

2.1 Raw Datasets

- The definitions of different types of telco data (CDRs, probe, and app-based data) are well written and clear.

- The significance and representativeness of app-based data from the MyVodafone app compared to the two other types of data could be elaborated. It is noted that the customer cohort is more limited in app-based data than the other data types, and that the customers using the app need to consent to the use of data. These two conditions (use of app and consent) presumably limit the number of customers whose app-based data is available for analysis.

2.2 Types of analysis

- 2.2.1 The use of NUTS3 regions as standard units for geographical location: The units can be relatively large in many countries, affecting the usability of data. In other words, telco data analysis on NUTS3 levels only reveals movement between regions, in many cases failing to identify movement patterns in smaller areas. However, at the same time the spatial inaccuracy improves privacy. Related to my general comment above, it would be interesting to elaborate on the balance between these two concerns.

- 2.2.1 Scaling of OD matrices to population levels: I appreciate the clear description of extrapolating the data to the whole population. One factor reducing the representativeness of telco data is that each operator is usually able to provide data only from their own customers, meaning that the data represents the operator’s customer base instead of the whole population. Does Vodafone’s customer base differ from the general population in terms of demography and was this acknowledged when extrapolating the data?

- 2.2.1 The rationale behind using the top three nighttime cells is not clear to me. Is this related to better accuracy of location (triangulation) or just a technical fact due to devices connecting to varying cells?

2.3 Privacy and Ethics

- The paper only refers to legal constraints in using telco data in the EU. While I understand that this is not a legal study, I think the ePrivacy Directive poses such fundamental restrictions to telecommunication data analytics in EU member states that it should be more clearly acknowledged also in this paper. For example, the anonymization and aggregation requirements are entirely based on law and significantly affect the accuracy of data.

- I appreciate the clear description of the anonymization efforts taken by Vodafone through aggregation of data. However, please see my general comment above about balancing anonymity and the usefulness of the data.

- Vulnerable demographic groups: The fact that even aggregated and anonymized telco data analysis can result in unintended consequences to minorities and other vulnerable groups is an interesting and important finding. On what criteria was the assessment of these vulnerable groups based? Is it possible to give examples of the kind of data that was subject to additional restrictions?

- In the second paragraph of this section you mention a Privacy Impact Assessment and in the next paragraph, a Group Privacy Impact Assessment, of which the latter is described as a novelty in responding to ethical concerns. I was confused as to whether these are two separate impact assessments.

4.2 Regulatory challenges and frameworks

- This section states that “the particular challenges related to full anonymization of this data are very well understood.” While the article touches on many of these challenges, I would take the opportunity to spell these challenges out in this section. Related to my general comment about the limitations of telco data, the easy identifiability of location data, the balance between accuracy and anonymity, and the legal and technical challenges of the anonymization process may presumably explain in part many of the choices Vodafone has made in determining group sizes and geographical areas, and the limitations to the availability of data. I believe addressing these choices more clearly would provide important background for the readers.

4.3 Data usage and usefulness

- This section mentions disaggregating self-isolation KPIs into gender and age groups. This sounds like an important factor for political decision-making that clearly improves the accuracy and usefulness of data. It would, however, be interesting to hear about the effect these additional attributes have on the aggregation of data, and the combination of the demographic information with telco data. Is the age/gender information derived from statistics about geographical areas? How were possible differences between Vodafone’s customer base and the general population considered?

- The list of countries in which the analysis was conducted (Italy, Spain, Portugal) could be mentioned earlier in the article for clarity.

5.1 Representativeness and limitations

- Population density and its effect on telco data analysis is explained well in this section. However, the topic feels somewhat out of place in the concluding section. Please see my above suggestion about a new, separate section on the limitations of telco data.

---

## [Reviewer Report]

*Comments to Author*: The paper describes Vodafone's efforts to provide mobile CDR and XDR based insights to several major stakeholders via data collaboratives during the management of the COVID crisis. The paper describes the datasets used for this purpose, gives examples of the indicators that are provided, briefly mentions privacy and ethics concerns, and provides a summary of existing initiatives.

While the main take home messages of the paper are not novel (e.g. that telco data can be very useful to provide insights to policy makers during the pandemic), the paper provides some insights into the existing data collaboratives with Vodafone. Examples of visualizations from the provided indicators are quite simple, and by themselves, just illustrate the potential of the data. What would be useful to provide, in addition to these examples, is a guideline for initiating new data collaboratives with Vodafone, or even a starting point for this, which is largely missing from the paper. The ethical and privacy issues are not discussed in detail, and raise more questions than they answer. This part of the paper can benefit from a more in-depth, detailed, and practical discussion.

Further comments, in order of appearance in the paper:

In Section 2.1, the definition of CDR includes data packets. A more typical definition includes voice and SMS in CDR, and data packets in XDR. Data packet exchange has a much higher frequency than voice/SMS, and it may be a good idea to make the reader aware of this. Subsequently, CDR is less "threatening" to privacy than XDR, because of its reduced frequency. If the first dataset includes data packets of applications running in the background, it will be better to call it XDR, instead of CDR.

In the beginning of Section 2.1, it will be useful to state the data collection period, and in particular, its relation to Covid measures. As the authors also note, pre-Covid and post-Covid mobility is different, and depending on stringency measures and lockdowns, show great variation across countries and time periods. Some variables, such as work locations of phone line owners, could be determined using pre-Covid data, and used during Covid, but it is more difficult to determine actual work location during lockdowns.

Section 2.2.2, it will be good to describe potential sources of bias in the estimates. Does the ratio of customers to population estimate also correct for market share of the particular mobile operator, as well as for children, who do not legally own phone lines? How recent are the population estimates from Eurostat, and do they include large movements of people over short periods, such as refugees?

Earlier CDR based collaboratives (e.g. Data for Development - Blondel, V. D., Esch, M., Chan, C., Clérot, F., Deville, P., Huens, E.,... & Ziemlicki, C. (2012). Data for development: the d4d challenge on mobile phone data. arXiv preprint arXiv:1210.0137.) have blurred base station locations, as these were considered to be sensitive information. Was there a similar processing in the calculations, and if yes, what is the impact of such processing in the results?

Section 2.2.3, please use the open form of KPI in its first occurrence.

Are the three potential locations pooled to describe the "home" location? Is this a good assumption? Were these three cells often bordering cells? Why not take the top location and pool the top three? What is the distribution of presence over the top, top two and top three locations? If only the top location is used, how does Figure 1 change?

In Section 2.2.4, the top cell is used instead of the top three. This makes sense, but the use of three cells in the previous subsection is not well-motivated. Are there social arguments to provide more insight there? Do many people have multiple homes?

In Section 2.2.4, Is the radius of Earth taken as 3671km? This small detail may help the reader. For the number of cells, do you use the list of unique cells, or does each visit count (possibly with home or other cell visits in between) as a different entry? This has an important implication. Suppose I visit 100 times a cell at a short distance, and once a cell at a large distance. In the case where unique cell list is used, the impact of the one visit to the large-distance cell will have a large impact on the radius of gyration calculation. The preference does not come clear in the text.

Figure 3 contains a large dip in collocation on 8-9 March. What is the reason for this dip? A similar but smaller dip is there in figure 4, on 8 March (and not 9th). Why?

In Section 2.3, the risks and risk mitigation steps are vaguely specified. " Therefore a country by country assessment was conducted": Who has conducted this assessment? Were they stakeholders from groups at risk involved? How are all the groups at risk identified?

"...where the risk was intolerable, the data analytics either was not conducted or additional aggregation controls were put in place to remove those potential insights.": Which locations had "intolerable risks" and why? Which aggregation controls were put in place for these specific cases?

This section is also not clear about whether the data (in its anonymized and aggregated form) left the company or not. It appears from "...all data is pseudonymised when processing the data into the anonymised reports in order to mitigate any privacy risks." that only the reports are shared, and not the data themselves. However, a few sentences later we have " Each data sharing activity - external to the company", which suggests that there are data sharing activities external to the company. It is not clear from the text what the shared data are, and with whom they are shared. An explicit list of shared items will be helpful to assess the privacy implications.

In Figure 5, it may be a good idea to superpose some stringency index or lockdown indicator markers on the data. Is the RoG increasing despite lockdown measures being in place? Or did the countries gradually relax the measures? The interpretation is quite different in both cases.

The quality of Figure 7 can be improved. The text is too small, the lines are too thin. A few simple guidelines can improve your visualizations very quickly. Check Edward Tufte's books on visualization for some useful principles of visualization. High quality plots can also be created easily with MATLAB, R, or similar software. The following link has some useful suggestions for instance: https://nl.mathworks.com/matlabcentral/fileexchange/35246-matlab-plot-gallery-publication-quality-graphics.

Section 4.1 is not sufficiently clear about whether such data collaboratives can be established at the moment with Vodafone, and what to do to initiate them. They are judged on a case-by-case basis, but are there guidelines for applying for a collaborative? Providing some contact points and guidelines will be useful.

Section 4.3 is a very useful summary on the existing data collaboratives. Which segmentation factors were used for these initiatives? Age and gender based disaggregation are mentioned for the study with IMF, but is this all? Were there no other factors involved in this and the other studies?

There exist other data collaboratives where different factors were involved (e.g. the Data for Refugees study involved a "refugee" tag in segmenting the CDR, and other indicators are computed internally by telecom operators. Salah, A. A., Pentland, A., Lepri, B., Letouzé, E., Vinck, P., de Montjoye, Y. A.,... & Dagdelen, O. (2018). Data for refugees: the D4R challenge on mobility of Syrian refugees in Turkey. arXiv preprint arXiv:1807.00523.). Each segmentation brings different concerns for the privacy and ethics review, but also offers additional insights. The Magicbox report cited in this section, for example, mentions disaggregating behavior according to poverty, where each (home) area is tagged with an indicator about average income.

The paper does not discuss some of the main concerns of mobile data sharing initiatives in times of humanitarian crises, such as pandemics. The short-term benefits are obvious, but what are the longer-term risks? Oliver et al. (2020) expresses some of these concerns:

"A key concern is that the pandemic is used to create and legitimize surveillance tools used by government and technology companies that are likely to persist beyond the emergency. Such tools and enhanced access to data may be used for purposes such as law enforcement by the government or hypertargeting by the private sector. Such an increase in government and industry power and the absence of checks and balance is harmful in any democratic state. The consequences may be even more devastating in less democratic states that routinely target and oppress minorities, vulnerable groups, and other populations of concern." (Oliver, N., Lepri, B., Sterly, H., Lambiotte, R., Deletaille, S., De Nadai, M.,... & Colizza, V. (2020). Mobile phone data for informing public health actions across the COVID-19 pandemic life cycle.)

An important item missing from the discussion (and stressed in the Oliver 2020 paper) is the potential of mobile CDR/XDR to provide real-time insights to the stakeholders. Data collection, aggregation, anonymization, and ethics clearances usually take a lot of time and effort, as the authors also stress. Subsequently, these analyses are typically conducted by looking at historical data. However, what is much more useful for policy makers is the real-time potential of such data. Mobile CDR can provide a snapshot of what is happening now in a country. Are any of the mentioned initiatives included such near-real-time insights? Is there an existing pipeline (including infrastructure, code, agreements, review and approval schemes, etc.), which can be rapidly activated to provide such timely insights?

---

## [Reviewer Report]

*Comments to Author*: This paper describes the experience of Telco company Vodafone in using mobile phone data to develop metrics to monitor mobility changes during the COVID-19 pandemic. Overall, the paper is interesting. However, there are several aspects that must be clarified and improved. I describe them in the following:

1) When describing the cell sites and the geographic information they convey, the authors say "[...] while still maintaining individual privacy as these locations are approximate and offer less resolution than GPS level datasets". While it is true that mobile phone data provide less precise geographic information then GPS datasets, individuals can be re-identified rather easily even in a mobile phone dataset. For example, de Montjoye et al. 2014 (https://www.nature.com/articles/srep01376) show that just four locations are enough to re-identify an individual in a mobile phone dataset, and more types of attacks are suggested and implemented in Pellungrini et al. 2017 (https://dl.acm.org/doi/10.1145/3106774). In general, the authors may refer to de Montjoye et al. 2018 (https://www.nature.com/articles/sdata2018286) for a summary of the privacy problems in mobile phone data and how to deal with them.

2) The authors use the term Call Detail Records (CDRs) to indicate records corresponding to "voice call, text message, multimedia message or data packet". In many papers, records that are created from data connection are usually referred to as eXtended Detail Records (XDRs). It would be useful to mention this ambiguity in the nomenclature in the paper (maybe the authors could say that they combine an individual’s CDRs and XDRs).

3) Although OD matrices are mentioned in Section 2.2, it is not clear how they are used to monitor mobility changes. It would be beneficial for the reader to briefly describe in which way OD matrices have been used by JRC. Do they just visualize their structure or extract relevant network metrics from them to track their evolution in time?

4) To be precise, the Radius Of Gyration (ROG) of an individual, as initially introduced by Gonzalez et al. 2008 (https://www.nature.com/articles/nature06958) for human mobility, is defined with respect to their centre of mass and not with respect to the individual’s home location. The authors should hence clarify that they use a variant of the original ROG. Moreover, as I understand from Figure 2, the authors computed an individual's ROG day by day, i.e., considering the phone records of that individual for that day only. In general, the ROG is used to describe individual mobility ranges for a wider temporal period (e.g., two weeks, see Gonzalez et al. 2008). Computing the ROG of an individual/day using records of the past week(s) could have provided a more reliable indication of the individual's mobility range. Maybe the authors could comment on that in the paper. Also, note that in the ROG equation variable n is not defined.

5) "For this 60 min buckets were processed throughout the day and looked into the number of customers seen in a specific cell in the network at the same time, as long as this was not their home cell (where it was assumed they would be in their houses)". The assumption that people are in their house when in the home cell is reasonable in densely populated areas (in which many towers are installed), but not very much in scarcely populated areas. Analogously, co-location in scarcely populated areas (where the coverage of a cell is large) is less reliable. These limitations related to the geographic distribution of towers should be mentioned here (and not only in Section 5) to highlight the limitations of the proposed metrics.

6) "In this case all the insights produced are aggregated at a level of 50 or more individuals in order to preserve individual privacy". I am not sure I understand what the authors do here. What does it mean that the data are aggregated at the level of 50 individuals? And why precisely 50?

7) "Whilst individual privacy could be preserved through anonymisation". As raised in point 1), anonymisation (if intended as the process of just removing the individual’s identities) is not enough to guarantee privacy preservation.

8) In the Data Availability Statement, the authors say that "Data and code are only shared on a strictly case by case basis as per Vodafone's Privacy and Security standards". Could you clarify what these standards are? For example, is it possible to access the list of ROGs of the individuals for a given day?

Minor points:

8) In general, the figures are hard to read and many of them are never mentioned in the main text. First, I would make all numbers and labels bigger; second, it would be useful to add country-specific information about salient events in the management of the pandemic (e.g., starting of lockdown periods, as vertical lines for example)

9) Many papers have been published in 2020 on using mobile phone data to monitor mobility changes due to non-pharmaceutical interventions to contrast the COVID-19 pandemic. It would be useful to mention the most relevant ones, at least for the countries mentioned in the paper (e.g., Italy and Spain).

---

## [Reviewer Report]

This work represents a summary of a global effort within Vodafone Group to tackle COVID-19 and provide relevant insights to institutions and policymakers. At a time of great changes in population behaviour, the insights provided by Vodafone have proven to be extremely relevant for many organisations and the impact these types of insights can have in policy making are great. At the same time and at the forefront of all of Vodafone's activities the ethical and privacy questions were addressed to ensure safety and trust from the public. We hope this paper sheds a light into how important these activities are and elicits more collaboration across the industry and public organisations to address societal problems.